# Modifying Orthobiological PRP Therapies Are Imperative for the Advancement of Treatment Outcomes in Musculoskeletal Pathologies

**DOI:** 10.3390/biomedicines10112933

**Published:** 2022-11-15

**Authors:** Peter A. Everts, Timothy Mazzola, Kenneth Mautner, Pietro S. Randelli, Luga Podesta

**Affiliations:** 1Education & Research Division, Gulf Coast Biologics, Fort Myers, FL 33916, USA; 2Breakthrough Regenerative Orthopedics, Boulder, CO 80305, USA; 3Department of Physical Medicine and Rehabilitation, Emory University, Atlanta, GA 30329, USA; 4Instituto Orthopedico Gaetano Pini, Milan University, 20122 Milan, Italy; 5Bluetail Medical Group, Naples, FL 34109, USA

**Keywords:** evolutionary medicine, autologous orthobiologics, platelet-rich plasma, platelets, angiogenesis, painkilling effects, immunomodulation, dosing, bioformulations, platelet–leukocyte interactions

## Abstract

Autologous biological cellular preparations have materialized as a growing area of medical advancement in interventional (orthopedic) practices and surgical interventions to provide an optimal tissue healing environment, particularly in tissues where standard healing is disrupted and repair and ultimately restoration of function is at risk. These cellular therapies are often referred to as orthobiologics and are derived from patient’s own tissues to prepare point of care platelet-rich plasma (PRP), bone marrow concentrate (BMC), and adipose tissue concentrate (ATC). Orthobiological preparations are biological materials comprised of a wide variety of cell populations, cytokines, growth factors, molecules, and signaling cells. They can modulate and influence many other resident cells after they have been administered in specific diseased microenvironments. Jointly, the various orthobiological cell preparations are proficient to counteract persistent inflammation, respond to catabolic reactions, and reinstate tissue homeostasis. Ultimately, precisely delivered orthobiologics with a proper dose and bioformulation will contribute to tissue repair. Progress has been made in understanding orthobiological technologies where the safety and relatively easy manipulation of orthobiological treatment tools has been demonstrated in clinical applications. Although more positive than negative patient outcome results have been registered in the literature, definitive and accepted standards to prepare specific cellular orthobiologics are still lacking. To promote significant and consistent clinical outcomes, we will present a review of methods for implementing dosing strategies, using bioformulations tailored to the pathoanatomic process of the tissue, and adopting variable preparation and injection volume policies. By optimizing the dose and specificity of orthobiologics, local cellular synergistic behavior will increase, potentially leading to better pain killing effects, effective immunomodulation, control of inflammation, and (neo) angiogenesis, ultimately contributing to functionally restored body movement patterns.

## 1. Introduction

The human body employs an endogenous method of cellular, and hence, tissue regeneration through stem- and progenitor cells, growth factors, signaling cells, cytokines, and other cell types that are present in almost every type of tissue. There is an unmet need in the advancement and development of innovative and less invasive treatment options for patients suffering from a variety of musculoskeletal (MSK) and spinal disorders. In recent years, several autologous biological cellular preparations, freshly prepared at the point of care, have emerged as a growing area of innovative medical health care settings, like interventional orthopedic procedures and surgical interventions. These autologous cellular materials are termed orthobiologics and include biological preparations like PRP, BMC, and ATC, delivering archetypal heterogeneous cell concentrates, have become an important biological therapeutical instrument. Platelets, mesenchymal stem cells (MSCs), platelet growth factors (PGFs), leukocytes, signaling cells, cytokines, proteinases, chemokines, interleukins (ILs), and others play potentially important roles by stimulating and enhancing tissue repair processes in orthobiological, regenerative, treatment plans [1].

PRP orthobiological applications show promising results and often meet the objectives of regenerative medicine therapy initiatives to assist the body in establishing new functional tissues to restore degenerative or defective ones. Furthermore, they provide therapeutic treatment for conditions where conventional therapies are inadequate or even inappropriate. PRP preparations and orthobiological treatment protocols have progressed vastly over the past two decades. Clinicians gained a better understanding of how PRP can affect cellular physiology and how it can influence patient outcomes.

Unfortunately, definitive and accepted standards to prepare different orthobiological bioformulation are still lacking [2,3,4]. Unmistakably, this non-consensus status quo we currently find ourselves in has contributed to some clinical studies with no positive, and on occasion negative, patient outcomes. Consequently, authors from these suboptimal designed or executed outcome studies do not recommend the use of orthobiologics for particular MSK pathologies [5,6,7,8].

It is our belief that changes in orthobiological preparations and injectates are imperative to generate more consistent orthobiologic products that can lead to more significant, consistent, and positive patient outcomes.

Overall, we argue that “one-size-fits-all” PRP orthobiological preparations and applications should be replaced by more nuanced and transformative approaches. These advances include the adoption of algorithms to determine cell dosing strategies specific to the pathoanatomic problem, as well as the use of physiologically different PRP bioformulations specific to the different tissues and pathologies being treated, in the same patient procedure. This calls for an unyielding awareness and understanding of the large variability in orthobiological products that are currently on the market, regarding cell type, quality, and quantity, and application volumes needed to achieve appropriate dosing.

We aim to discuss the physiological variability of orthobiological preparations and specific bioformulations regarding their efficacy in immunomodulation, angiogenesis, pain downregulation, and tissue repair. Additionally, we address recent advances in platelet dosing, the specific roles of particular leukocytes, and the platelet-leukocyte interactions [3]. Ultimately, a better understanding of orthobiological cellular behavior in fascia, ligaments, tendons, cartilage, bone, muscle fibers, and nervous filaments, will contribute to improved mechano-metabolic conditions to restore MSK function after injury or tissue degenerative processes, as many interconnecting tissue structures are often involved [9]. It is only fair to assume that within the same functional orthopedic target (e.g., knee joint), different tissue structures and pathologies would respond differently to different orthobiological treatments in facilitating specific tissue repair processes.

In this review, we discuss and recommend transformations in routinely accepted orthobiological PRP preparations and bioformulations. Founded on the principles of evolutionary medicine [10], an increased awareness and understanding in ortho-biology is discussed to reevaluate the multifaceted tissue repair responses and evidence-based PRP strategies in the management of orthopedic and sports medicine pathologies.

## 2. Background Details Platelet-Rich Plasma

Autologous PRP is a centrifugated and processed liquid fraction of harvested fresh peripheral blood, characterized as a heterogenous and complex composition of multi-cellular components in a small volume of plasma, with a significant increase in platelets compared to baseline values, as they should be the primary cells in any PRP preparation [3]. A notable advantage of PRP is its autologous nature with some devices presenting ingenious, high quality, preparation techniques. Most importantly, PRP applications are safe, with no known systemic adverse effects, other than pain and swelling post treatment, when compared to other non-autologous biologics [11,12].

PRP therapies have been used for many different indications for more than 30 years, with clinical benefits and promising patient outcomes [13,14]. Unfortunately, despite global interest, PRP utilization is often overshadowed as there is no consensus on standardization with accepted preparation standards [15]. This interest stimulated the introduction of many devices with currently more than 50 different PRP, and PRP-like, systems commercially available and using different preparation protocols [4]. Not surprisingly, extreme variability in PRP platelet concentrations, cellular properties, and bioformulations has resulted. These tremendous PRP differences have caused challenges in establishing consistent clinical effectiveness, practicality, and growing clinical treatment needs. Not surprisingly, these profound dissimilarities likely contribute to the inconsistencies in patient outcomes and recommendations, as demonstrated in several systematic reviews, meta-analyses, and randomized controlled trials [3]. Fortunately, proteomics-based techniques and profiling [16,17] have helped elucidate PRP’s cellular functions and its effects on treatment outcomes. Regrettably, there is no consensus on uniform PRP preparation techniques, let alone the subsequent use of orthobiological application strategies regarding platelet dosing and employment of different bioformulation. Ideally, PRP utilization should follow a clinical PRP (C-PRP) recipe, imaginably contributing to substantial tissue repair mechanisms, eliciting significant positive clinical outcomes.

### 2.1. Platelet-Rich Plasma Properties

Based on the underlying scientific reasoning for PRP therapy, an injection of platelets may initiate tissue repair through restorative, regenerative pathways by releasing many biologically active growth factors, cytokines, lysosomes, and molecules as well as adhesion proteins.

#### 2.1.1. The Platelet Secretome

Proteomic based analysis techniques created more awareness regarding the importance of the platelet secretome constituents following platelet activation [18]. These analysis techniques revealed the functional importance of the innumerable cellular activities of the various platelet constituents present in the storage granules [19]. More specifically, an increased understanding of the platelet secretome’s biological and cellular functions and mechanisms in tissue repair pathways following PRP applications will instigate more effective PRP treatments with improved clinical outcomes. Therefore, different pathologies with their own unique cellular composites and tissue pathoanatomic conditions should each be treated with unique autologous clinical PRP products that meet their specific bio-restorative and cellular dosage requirements.

#### 2.1.2. Platelets

Platelets are small, anucleate, discoid blood cells (1–3 μm), and under normal circumstances their in-vivo half-life is seven days. In adults, the average platelet concentration ranges from 150 to 350 × 10^6^/μL of circulating blood. Platelets are created in the red bone marrow, where they are pinched off from their hematopoietic progenitor cells, megakaryocytes. Thereafter, they are released, in a non-activating state platelets into the peripheral circulation on an on-going basis [20]. Inside the platelet, three different intra-platelet structures subsist: α-granules, dense granules, and lysosomes [21]. The outer platelet cell membrane is covered with glycoprotein receptors and adhesion molecules. Following both passive and active activation processes, the three platelet structures release their content.

#### 2.1.3. Platelet α-Granules

In the early years of PRP use, α-granules were the most mentioned intra-platelet structure because they contained numerous PGFs, as shown in Figure 1. To a lesser degree, regulators of angiogenesis and coagulation factor are also found in α-granules [22].

As part of the platelet secretome, a number of additional factors, chemokines, and cytokines are also secreted, including platelet factor 4, pro-platelet basic protein, P-selectins (integrin activators), and RANTESs (Regulated by T-Cell activation and probably secreted by T-Cells). They function primarily as recruitment and activation agents for other immune cells, or as initiators of inflammation of endothelial cells [23].

#### 2.1.4. Platelet Dense-Granule Constituents

The second intra-cellular platelet structures are the dense granules. These substances act by converting platelet activation and thrombus formation, including histamine, ADP polyphosphates, 5-HT, and epinephrine [25]. It is also worth noting that several constituents of platelet cells are capable of modulating immune system cells [26]. Particularly, platelet ADP is identified by dendritic cells (DCs), leading to increased antigen endocytosis. DCs are vital cells in T-cell immune responses and navigate the protective immune response by linking the innate and adaptive immune systems [25]. T-cell migration and differentiation of monocytes into DCs are stimulated by other platelet dense granule constituents (glutamate and 5-HT) [26]. Potentially, C-PRP has considerable immune regulatory and modifying effects, as these dense granule-derived immune modifiers are significantly increased in concentrated PRP preparations.

#### 2.1.5. Platelet Lysosomes

Several studies have shown that following platelet activation, α- and dense granule secretion takes place in vivo in humans. Only very few data are available on the in vivo release of lysosomal content, which contain an array of acid hydrolases, as reflected in Figure 1. In general, lysosomal functions have not been well studied, although Heijnen and van der Sluis cited their roles as contributors to the digestive system of the cell, serving both to destroy material taken up from outside the cell and to digest archaic cytosolic components. Furthermore, they mentioned lysosomal activities in extracellular functions, such as fibrinolysis, supportive in vasculature remodeling, and the degradation of extracellular matrix components [27]. In this regard, lysosome proteases are alleged to play a role in tendon homeostatic mechanisms as they potentially can cleave the tendon ECM [28]. Interestingly, as lysosomes contain proteases and cationic proteins with bactericidal activity [29], they are interconnected with macrophages in phagocytosis [30].

## 3. Clinical PRP Recipe Considerations

It is obvious that in-vitro and animal PRP trials and procedures and protocols are not perfect settings to translate these outcome findings into clinical practice. Similarly, PRP product comparison studies are often not informative to support decision making processes, as these studies show a large difference in platelet concentrations and cellularity [15]. PRP specimens are living biomaterials and their biology is as complex as blood itself and likely more complex than traditional pharmaceutical drugs, especially after minimal manipulative centrifugation procedures to concentrate blood cells. Consequently, the ratios of the cells in PRP are inevitably changed, which is seldomly discussed with respect to patient outcomes. Notably, patient outcomes following clinical PRP procedures, are reliant on the intrinsic, versatile, and adaptive characteristics of the patient’s blood, including innumerable other cells that can be present in the PRP specimen [31] and the interaction with the recipient local microenvironment which is influenced by the overall health of the individual being treated.

At present, C-PRP formulations have been shown to lead to significant positive patient outcomes when used in numerous pathologies [32,33,34,35,36]. C-PRP is a complex composition of autologous concentrated platelets, with the option of adding or removing other cellular components, in a small volume of plasma, acquired from a fraction of centrifuged peripheral blood.

### 3.1. Platelets, Platelet Concentrations, Total Deliverable Platelets, Platelet Dosing

The therapeutic actions of PRP are based on the release of a large quantity of platelet factors implicated in tissue repair and regeneration, following platelet activation. The formed platelet plug functions as a temporary fibrin scaffold, allowing cells to proliferate and differentiate [37].

It is fair to assume that PRP preparations with higher platelet concentrations will generate an elevated local concentration of released platelet bioactive factors. However, this is only true if the injected PRP volume is also taken into consideration. Commonly, the increase in platelet concentration over baseline values or the absolute PRP platelet concentration are exploited as quality indicators for injected PRP. However, this is only part of the story as these parameters only inform physicians about the performance of PRP devices and are not sufficient as qualitative and quantitative PRP treatment parameters. Succinctly, the concentration of platelets or multiples of platelets above baseline alone do not accurately describe the total number of platelets (platelet dose) delivered to tissues. Platelet dosing should be calculated by multiplying the PRP volume that is injected in one treatment site by the known number of platelets per volume (platelet concentration). Multiplying the actual platelet concentration by the injected PRP volume reflects the total delivered platelets (TDP) to a treatment site and is synonymous with the term “platelet dose”. Thus, the differences in the actual “platelet dose” between PRP devices can be immense. This is in contradistinction to when only platelet concentrations or platelet increases above baseline are referenced as the quality parameters of injected PRP. In Table 1, the differences in TDP as platelet dose are clearly demonstrated in a same patient model using different PRP systems. Nevertheless, there are marked differences in baseline platelet counts between patients, making it difficult to determine the exact correlation between platelet dose and released platelet bioactive growth factors and agents [38]. Importantly, different PRP preparation devices produce different platelet capture rates, affecting the TDP potential [39,40]. Several in-vitro PRP studies have demonstrated that cells respond to PRP in a platelet dose-dependent manner [41,42].

### 3.2. Platelet Dose as a New Quality Indicator for PRP Orthobiologics

At present, clinicians should consider adopting TDP as a new and better PRP quality parameter, as this delineates how many platelets are accurately delivered to a single treatment site. Adopting this new quality treatment standard for PRP orthobiologics is supported by the translation of in-vitro data to clinical practice. Giusti et al. clearly demonstrated the need to deliver 1.5 × 10^6^ platelets/μL to induce a significant angiogenic response [43]. Per Giusti’s study, in order to induce angiogenesis, a 7 mL C-PRP treatment vial should therefore contain 10.5 × 10^9^ TDP.

Further studies are warranted to understand what TDP dosing is optimal for specific tissue types (tendons, ligaments, and cartilage) and conditions and MSK pathologies (tendinopathy vs. tears), and the chronicity of the disorder (acute vs. chronic).

Similarly, Berger et al. indicated that higher platelet concentrations, prepared from platelet lysate preparations, resulted in more tenocyte proliferation and migration, in a dose dependent manner [35]. More recently, Bansal and associates injected PRP in patients diagnosed with knee OA with 10 billion TDP [33]. They observed a consistent clinical effect regarding function, pain, and inflammatory markers after a single injection over a 12-month period. Additionally, other MSK oriented PRP studies have identified platelet dose and bioformulations as key characteristics that contribute to consistent analgesic effects [34,44].

We believe that C-PRP should be characterized by its absolute platelet concentration to optimize platelet dosing strategies, rather than continuing the earlier characterizations of PRP as a platelet concentration, or multiple above baseline values [37]. Therefore, we recommend that C-PRP have a minimum platelet concentration of more than 1 × 10^6^/μL [45] and that the volume required to reach the optimal platelet dose be accounted for. Clearly, any prepared C-PRP volume should be based on a patient-specific and individualized treatment plan. This treatment plan includes the total number of treatment sites and the injection volume per treatment site needed to achieve appropriate platelet dose. This information should be used to calculate the pre-donation volume required to produce injectates with a specific platelet dose, based on the patient’s pre-procedure baseline whole blood platelet concentration.

#### 3.2.1. PRP Platelet Dose and Painkilling Are Interrelated

In 2008, Everts et al. [46] were the first to publish on the analgesic effects of an activated PRP formulation in patients undergoing shoulder decompression.

A significant reduction in visual analog scale scores for pain and use of opioid pain medication, as well as a more successful post-surgical rehabilitation were observed. A variety of MSK oriented clinical studies have demonstrated a significant reduction, or even elimination of pain after PRP treatments [44,47]. However, others reported little to no pain relief [48,49]. Not surprisingly, platelet dosing and the PRP bioformulations have been identified as key features that contribute to consistent painkilling effects [33,50]. Furthermore, the type of injury, specific tissue types, PRP delivery route and application techniques, and the use of platelet activation have been mentioned as variables affecting pain modulation [51]. Notably, Kuffler studied the use of PRP to alleviate pain in patients suffering from chronic neuropathic pain, tributary to damaged non-regenerated nerves [52]. In PRP treated patients, the neuropathic pain started to decrease within three weeks. After that period, pain was eliminated or significantly reduced for more than six years.

Similar painkilling effects were observed by Mohammadi et al. in post-surgical wound care patients [53], observing a higher incidence of angiogenesis in patients treated with PRP. They concluded that the physiological aspects of wound pain were related to vascular injury and skin tissue hypoxia. Therefore, it was concluded that the reinstitution of neo-angiogenesis was mandatory to optimize tissue oxygenation and nutrient delivery. Likewise, Johal and colleagues concluded in a systematic review of PRP use in orthopedic indications, that PRP applications consistently contributed to pain reduction [54]. Noteworthy, intradiscal PRP injections led to the conclusion that pain killing was significantly correlated with PRP platelet concentrations [50], where higher platelet counts (>1.0 × 10^6^/μL), triggered a more promising painkilling response. These findings are in agreement with the findings Lutz and associates [34]. They determined that a high dose of platelets (>10 times baseline platelet level) significantly reduced pain following intradiscal injections. Remarkably, a significantly higher patient satisfaction rate was observed in patients who received a higher platelet dose. Currently, the optimal PRP platelet dose and bioformulation offering maximal pain relief is unknown. However, published data suggests that PRP should contain at least 1.0 × 10^6^/μL platelets to provoke pain killing effects [50]. Intriguingly, Yoshida et al. demonstrated complete pain relief in a rat model when the PRP platelet concentration was 1.0 × 10^6^/μL, whereas half this concentration incited significantly less pain reduction [55].

#### 3.2.2. Suggested Mechanisms Causing Painkilling

The exact mechanisms behind the painkilling effects of PRP have not yet been fully elucidated. Everts et al. postulated that the platelet dense granules release high concentrations of 5-HT in activated platelets. They hypothesized a mechanistic role of 5-HT in painkilling. The C-PRP platelet concentration is 5–7-fold higher than in peripheral blood, implicating that the potential to release platelet 5-HT is astronomical, as confirmed by Sprott et al. [56]. Concisely, in response to tissue injury or surgical trauma, platelets, mast cells, and endothelial cells release endogenous 5-HT [57], interfering with peripheral and multiple neuronal 5-HT receptors, potentially decreasing the extent of pain following noxious stimulation [58]. Another possible analgesic mechanism cited is a decrease in tissue inflammation followed by tissue repair. Araya et al. observed in an experimental arthritis model, using PurePRP, a reduction of pain-related behavior and inhibition of synovial inflammation [59].

### 3.3. Contributions of Leukocytes in C-PRP

In the orthobiological literature, unfortunately most studies do not include detailed PRP preparation methods, although they are recommended for study reproducibility. This lack of adequate reporting is surprising because PRP biological preparations contain widely varying leukocyte populations and concentrations, which can significantly contribute to pro-inflammation, immunomodulation, nociceptive effects, tissue repair, and tissue regeneration. In general, PRP preparation protocols are highly underreported with inconsistent laboratory data reporting of platelet counts, platelet dose, and specific leukocyte population concentrations.

Unfortunately, due to their unique design characteristics, the various PRP preparation devices produce very different leukocyte counts and populations with dissimilar neutrophil, lymphocyte, and monocyte cell concentrations and cell ratios in the final PRP preparation [60]. The presence of leukocytes in PRP preparations is device manufacturer and preparation protocol dependent. In general, plasma test tube-based PRP devices produce very few leukocytes and significantly less platelets when compared to more advanced PRP devices. More sophisticated PRP devices produce a buffy coat stratum where leukocytes are significantly concentrated [61], except for the eosinophils and basophils, as their cell membranes are too fragile to withstand the centrifugal processing forces. Even then, these buffy coat devices utilize different two-step preparation protocols. More specifically, only a very few devices can pellet and resuspend platelets with the plasma supernatant, thus creating either leukocyte-poor PRP (LP-PRP) or leukocyte-rich PRP (LR-PRP) [1].

Physicians should carefully consider the inclusion of leukocytes in C-PRP bioformulations, as these cells influence the intrinsic biology in both acute and chronic tissue lesions due to their immune and host-defense mechanisms. Hence, the presence of specific leukocyte populations in C-PRP preparations will likely lead to significant differences in cellular and tissue effects. Considerations regarding C-PRP bioformulations should thus be based on the specific MSK pathology being treated, the cellular properties of tissue structures, and the chronicity of the tissue pathology.

#### 3.3.1. Lymphocytes

A buffy coat, double spin LR-PRP preparation typically contains more lymphocytes than neutrophils or monocytes, with lymphocytes dominating the leukocytic population [62]. There is a greater concentration of mononuclear T and B lymphocytes in buffy coat PRP preparations than in other leukocyte types. In cell-mediated cytotoxic immunity, they play a crucial role. The cytokines interferon-γ (IFN-γ) and interleukin-4 (IL-4) further enhance macrophage polarization [63]. Based on their study of a mouse model, Weirather et al. demonstrated that regular T lymphocytes modulated the differentiation of monocytes and macrophages, subsequently influencing non-inflammatory cellular responses at the site of injury [64].

#### 3.3.2. Neutrophils

Neutrophils are essential leukocytes and join in numerous healing pathways, creating dense barriers against invading pathogens [65] in conjunction with platelet anti-microbial peptides [66]. The presence or absence of concentrated neutrophils in PRP products is controversial, but their inclusion should be contingent on the interventional treatment objectives. Exacerbated tissue inflammatory levels requiring neutrophils can be necessary in PRP biological applications for chronic wound care [67], or for applications related to the stimulation of bone growth or bone healing [68]. Often, LR-PRP is recommended as an orthobiologic for chronic tendinopathy [69,70]. Furthermore, neutrophils have been shown to function in additional ways, such as in angiogenesis and tissue regeneration, in several recent studies [65].

Jones et al. have also described the importance of neutrophils in the resolution of tissue inflammation thus leading to tissue repair via Annexin A1, lipid mediators, IL-10, bioactive microparticles and neutrophil apoptosis [71]. Notably, Ozel et al. discussed the direct pro-angiogenic function of neutrophils and their capability to activate pro-angiogenic functions of other immune cells and, thus indirectly contribute to angiogenesis [72]. However, neutrophils are also linked to possibly harmful effects and might not be indicated for some treatments, as suggested by Zhou et al. in an in-vitro study [73]. Other neutrophil-mediated adverse properties are the acute release of inflammatory cytokines and MMPs, promoting pro-inflammatory and catabolic effects when PRP is discharged to various tissue types [74]. Finally, several systematic reviews have branded LP-PRP as the preferred PRP formulation to treat joint OA [75]. However, Lana and co-workers have opposed this theory, suggesting that PRP leukocytes release of both pro and ultimately anti-inflammatory molecules, therefore they hold pivotal roles in the inflammatory processes. In particular, the combination of neutrophils and activated platelets could have a more positive than detrimental effect on tissue repair in OA PRP treatments. Additionally, they elaborated on the importance of the plasticity of monocytes and the roles of their phenotypes in both non-inflammatory and tissue repair [76].

#### 3.3.3. Monocytes

Monocytes are heterogeneous cell populations and originate from hematopoietic progenitor cells in the bone marrow and their phenotypic manifestation and regenerative capabilities play key roles in immunomodulatory processes and tissue repair mechanisms [77]. In PRP treatment devices, they can be prominently present or absent depending on the PRP device used.

In their native state, depending on local microenvironmental stimuli, like inflammation, circulating monocytes leave the bloodstream and are recruited to injured or degenerated tissues via homeostatic mechanisms. Monocytes act as progenitor cells for macrophages, and together with dendritic cells they embody the mononuclear phagocyte system (MPS) [78]. Apoptotic or necrotic cells, macrophages, local growth factors, and pro-inflammatory cytokines are all responsible for triggering the differentiation of monocytes into MPS cells in degenerating tissues [79]. In a diseased microenvironment, monocyte rich C-PRP is likely to stimulate major cellular changes, as monocytes differentiate into macrophages that become macrophage phenotype-1 cells (MP-1) [77]. This MP-1 cell is characterized by inflammatory cytokine IFN-γ secretion and nitric oxide production, resulting in its effective pathogen killing mechanisms along with the production of VEGF and FGF. The macrophage phenotype-2 (MP-2) is characterized by its tissue repair anti-inflammatory cells, producing mainly extracellular matrix components, the anti-inflammatory cytokine IL-10, and angiogenic factors [78]. Remarkably, the pro-inflammatory MP-1 can switch to the pro-repair MP-2, based on local environmental signaling and perhaps can be influenced by the type of PRP used [80]. From these data, it is reasonable to assume that C-PRP preparations containing a high concentration of monocytes, and ultimately upregulating MP-2 tissue macrophage phenotypes, are likely to contribute to enhanced tissue repair processes, because of their non-inflammatory tissue repair and cell signaling capabilities.

### 3.4. No Beneficial Roles for Erythrocytes in C-PRP

No beneficial function of erythrocytes or red blood cells (RBCs) in tissue regeneration has been established. RBCs consist of protein-bound heme and iron molecules, responsible for transporting oxygen to tissues and binding and transporting tissue bound carbon dioxide to be removed by the lungs [81].

Under conditions of shear forces (e.g., immune-mediated processes, oxidative stress, during deficient phlebotomy procedures, and inadequate PRP protocol handling, high shear forces), the RBC cell membranes will disintegrate, with the subsequent release of toxic hemoglobin, measured as plasma-free hemoglobin (PFH)), hemin, and iron [82]. These RBC degradation products together lead to detrimental and cytotoxic tissue effects, causing oxidative stress, loss of nitric oxide, activation of inflammatory pathways, and immunosuppression. Finally, these cytotoxic properties lead to microcirculatory dysfunction, local vasoconstriction, and lead ultimately to significant tissue injury [83]. Additionally, when PRP containing RBCs are delivered to treatment sites, a local response called eryptosis might occur. As a consequence of eryptosis, macrophage migration inhibitory factor (MIF) is discharged from RBCs [84], ultimately leading to several significant physiological phenomena. The potent, multifaced, inflammatory cytokine MIF impedes the migration of monocytes and macrophages into injected tissues and hinders the migration of stem cells and fibroblast proliferation. Adequate C-PRP centrifugation and preparation processes thus must aim to eliminate, or minimize, the presence of RBCs [85].

### 3.5. Variables in High Dose Multicellular C-PRP

Lacking consensus on standardization of PRP formulations has contributed to the development of a wide variety of different orthobiologic devices with inconsistent and dissimilar cell capture rates, with varying platelet and other non-platelet cellular constituents [31,86,87]. Therefore, practitioners should understand the significant preparation variables required to produce a high-yield multicellular PRP specimen. The quality and quantity of specific PRP products, as determined by platelet dosages and leukocyte subpopulation bioformulations, are directly related to high cellular recovery rates from a unit of whole blood after gravitational density separation. Moreover, these advanced PRP devices should be proficient in preparing different validated bioformulations to produce various and specific platelet dosing and leukocyte populations and concentrations.

Nonetheless, the extensive variations in reported PRP preparation protocols, such as the total donated whole blood volume, type of anticoagulants used, device physical characteristics, platelet and leukocyte recovery rates, and centrifugal variances, make it very difficult to compare PRP outcome results accurately [15]. These variable preparation characteristics, responsible for the heterogeneity of PRP preparations in MSK applications, have been discussed in more detail by Cherian et al. [88].

The PRP centrifuge performance variables, like number of spin cycles, centrifugation speed and duration per spin cycle, and the acceleration and de-acceleration speed, are rarely mentioned. Thus, the characteristics of the ultimately extracted orthobiological products are scarcely understood. Piao et al. nicely defined that the critical factors to guarantee effective cellular yields in PRP orthobiologics are (1) centrifuge acceleration profiles and (2) spin time for the maximum recovery rate of platelets and leukocytes. Other substantial factors that determine the total number of available platelets for dosing are the total pre-donated anticoagulated blood volume prior to PRP processing, and the geometrical mathematics and physical properties of PRP devices used for cell concentration [89].

## 4. PRP Preparation Methods

The literature specifies that optimal PRP preparations are generated by so-called double-spin protocols, as these systems are capable of creating a layered buffy coat stratum [31].

### 4.1. Single Spin vs. Double Spin Preparation Devices

The literature confirms that optimal PRP preparations are generated by so-called double-spin protocols, as these systems are capable of creating a layered buffy coat stratum 28. This is in contrast with single-spin test tube-like “PRP” devices that prepare a product from the acellular plasma layer, excluding all leukocytes and erythrocytes [89]. They typically have a very low platelet capture rate when compared to the more advanced double-spin devices [61]. The differences between single and double spin PRP characteristics included a profuse variability in cellular compositions and platelet dose [90]. Marques et al. correlated inferior PRP treatment outcomes with poor quality and inconsistent PRP products [91]. Fadadu et al. confirmed [4] that most single spin test tube type “PRP” systems produced platelet counts of 0.52 times baseline values, effectively producing platelet poor plasma. In contrast, they stated that double-spin PRP devices generate significantly higher platelet concentrations [4]. Remarkably, Magalon and co-workers noticed a large heterogeneity among PRP devices, concluding that the total number of PRP platelets (i.e., the platelet dose) is directly correlated to the pre-donated whole blood volume, the PRP device centrifugal forces, and preparation protocols [2].

It’s indisputable that extensive differences in PRP properties, platelet dose, and bioformulations between single and double-spin PRP devices are directly related to profound variances in product design concepts, centrifugation parameters, whole blood predonation volumes, PRP preparation protocols, and finally PRP volumes. Predictably, these major differences in PRP quality and treatment volumes likely lead to significant variations in patient outcomes [6,7].

### 4.2. Cellular Gravitational Density Separation Principle to Produce Orthobiologics

Swing-out, horizontal centrifuges are most frequently used at the point-of -care to process a unit of whole blood as they allow for better, and more equal, separation of whole blood cellular and plasma components based on the densities of those components.

Blood in PRP preparation devices is processed by employing PRP centrifuges, acting on centrifugal forces in the radial direction, gravitational forces in the downward direction, and drag forces in the opposing direction of cellular motion. The differences in physiological blood cell sizes and densities represent the ultimate driving force accounting for the whole blood separation process [92]. In addition, horizontal centrifuges lessen the level of cell trauma when compared to angulated centrifuges, improving the quality of the PRP [93]. Laboratory validated double-spin PRP preparation protocols use programmed centrifugal settings to enable adequate g-forces and processing time settings to carefully exfoliate the whole blood cellular components, as displayed in Figure 2. The magnification in C clearly shows the presence of the multicellular buffy coat stratum. Following cellular gravitational density separation, cells are accumulated at the bottom of the PRP concentration device, where platelets have the lowest density. An organized multicomponent, greyish, buffy coat layer with minimal RBC content is exposed, illuminated in D.

## 5. Mimetic Properties of PRP in Tissue Repair

The principal advantages of PRP preparations comprise of its autologous nature and intelligent preparation techniques in double spin devices. Most importantly, the literature indicates that the use of autologous PRP products is safe, with no known systemic adverse effects, compared to other non-autologous biologics [12]. Tissue repair with PRP following degenerative processes or trauma is a complex biological process whereby platelets act as a natural reservoir for PGFs, proteins, cells, and molecules as shown in Figure 1. The final repair outcome should be the restoration of tissue integrity complemented by improved function. Physiologically, the healing cascade obeys to diverse phases, based on a variety of uninterrupted cellular activities and extracellular signaling processes. Following active, or in situ, PRP platelet activation, platelets will release their granular content, lysosomes, chemokines, and other platelet cytokines. Among others, an abundance of PGFs, platelet adhesion molecules, and 5-HT are released. Successively, a variety of cellular and molecular interactions are initiated in the treated area, including chemotaxis, cell adhesion, migration, and cell differentiation. Initially, PRP technologies were introduced to mimic the initiation of bone healing in intraoperative orthopedic, trauma, and maxillo-facial surgical procedures [46]. A decade later, PRP applications emerged in interventional, non-surgical procedures to support healing cascades over a broad range of applications [95]. In order to mimic nature’s healing cascades, PRP platelet proteins and molecules should be capable of inciting (neo) angiogenesis and immunomodulatory activities by stimulating cell proliferation, adding to chemotactic cell migration, and expediting the activities of mesenchymal and neurotrophic factors [96,97]. Combined with the presence of high concentrations of leukocytes, autologous multi-cellular PRP preparations are capable to jumpstart the full complexity of biological cell signaling, including microenvironmental signaling required to regulate biological mechanisms that lead to tissue restoration and integrity [3].

### 5.1. PRP Platelets and Angiogenetic Properties

Angiogenesis is a vibrant, multistep process with numerous biological mechanisms, including endothelial cell migration, proliferation, differentiation, and cell division. Adequate angiogenesis, with the sprouting and organization of micro-vessels from pre-existing blood vessels, is mandatory in orthobiological tissue repair processes, as in diseased and degenerative tissues, the milieu in these microenvironments is characterized by low oxygen tension, low pH, and high lactate levels [3]. These tissues respond accordingly by restoring angiogenic functions with local angiogenic factors. Platelet pro- and anti-angiogenic factors modulate angiogenic activities following PRP treatment in these MSK pathologies, while maintaining a physiological combination and ratio of these factors, which are essential for the formation of long-term functional blood vessels [98]. The angiogenesis model of tissue repair is an integrated part of the complex classical healing cascade. This quirky angiogenetic cascade is often overlooked, although this is an essential contributor and pathway in tissue reparative processes, with decisive roles for PRP platelets. This is of substantial clinical importance, as platelets intercede with angiogenetic activities, stimulating (neo) angiogenesis to create vascular architecture. Restoration of blood flow in defective tissues will safeguarding the high metabolic activity of tissue reparative processes, as new blood vessels can deliver oxygen and nutrients while simultaneously removing catabolic byproducts [53]. The activities of the angiogenesis cascade are characterized by platelet growth factors to cell membrane-receptors, cell–cell communication, and cell–matrix interactions, regulated by the balance between pro -and anti-angiogenetic platelet factors [99]. Ultimately, these processes contribute to the healing in areas of poor vascularization, such as meniscal tears, tendon injuries, and other areas with poor vascularity, as mentioned by Landsdown and Fortier [100].

Notably, the delivery of PRP with a high platelet dose, containing high concentrations of the stimulatory pro-angiogenic PGF VEGF, induces more powerful angiogenesis, vasculogenesis, and arteriogenesis through stromal cell-derived factor-1a binding to endothelial progenitor cell receptors. Another important and essential factor in restoring angiogenic pathways is the synergy between multiple PGFs. Based on the synergistic activities of the growth factors PDGF-BB and VEGF, Richardson et al. demonstrated an increase in the rate of formation of a mature vascular network in comparison to that of the individual factors [101]. It is important to mention that Giusti and co-workers concluded in a dose-defining study that 1.5 × 10^6^ platelets/μL was the optimal platelet dose for promoting angiogenesis [43]. In contrast to lower platelet preparations with less than 1.5 × 10^6^ platelets/μL, PRP products with a corresponding platelet dose likely contribute to significant positive angiogenetic effects.

### 5.2. PRP Leukocytes and Immunomodulatory Properties

The human body can readily identify foreign bodies and injured tissues in both acute and chronic conditions. This is the basis for initiation of inflammatory pathways and is related to start of the wound healing cascade. We recognize both the innate and adaptive immune systems, where leukocytes hold essential roles in both systems and join in the overlap between both. Particularly, monocytes, macrophages, neutrophils, dendritic cells, and natural killer cells have fundamental tasks in the innate system, to nonspecifically identify intruding microbes or tissue fragments and stimulate their clearance. However, lymphocytes and their subsets have similar capacities in the adaptive immune system [79]. Under homeostatic circumstances, platelets are among the first cells to identify endothelial tissue injury and detect the presence of microbial pathogens. Following PRP injections, high concentrations of platelets accumulate at treated tissue sites where they aggregate and as the platelets become activated, they in turn release many platelet agonists followed by an abundance of platelet biomolecules. This enhances further platelet activation while expressing platelet chemokine receptors. Under normal circumstances, this will result in a rapid accumulation of peripheral platelets at the site of injury or infection [102]. Subsequently, neutrophils, monocytes, and dendritic cells are recruited for an early-phase immune response. As orthobiological PRP therapies provide inflammatory stimuli, platelet receptors change their surface expression to stimulate platelet–leukocyte interactions by forming platelet–leukocyte aggregates to regulate inflammation and ultimately tissue repair. More precisely, neutrophils and monocytes are both active participants in these aggregate formations, thereby contributing to the innate immune response [103].

#### 5.2.1. Platelet-Neutrophil Interactions

The platelet interaction with neutrophils is central to initiating the immune response. Platelet–neutrophil interactions are mediated by platelet adhesive molecules P-Selectin [104], eventually releasing reactive oxygen species (ROS) and myeloperoxidase (MPO) from neutrophils [65], a phenomenon known as leukocyte oxidative burst. Additionally, platelet-neutrophil interactions lead to neutrophil cellular degranulation with the development of neutrophil-extracellular traps (NETs), a pathogen killing neutrophil mechanism which traps bacteria and kills them by a process termed NETosis [105].

#### 5.2.2. Platelet-Monocyte Interactions

C-PRP activated platelets interact and modulate monocyte functions during inflammatory or infectious stimuli. Monocytes migrate and adhere to diseased and degenerative tissue, secreting inflammatory molecules that are capable of modifying chemotaxis and contribute to proteolytic properties [106]. Furthermore, after C-PRP delivery, deposited platelets modulate the effector functions of monocytes, activating and differentiating several immune cells, facilitating the endogenous monocyte oxidative burst. Assuredly, they differentiate into several macrophage phenotypes [107]. Moreover, platelets stimulate monocyte and macrophage responses in the development of dendritic and natural killer cells, causing T and B lymphocytes responses, with specific roles for the various T helper (Th) cells [108].

### 5.3. Platelet-Leukocyte Interactions in PRP Applications

In the various available PRP preparations, leukocytes can be either included or avoided from treatment specimens. In the literature the presence of leukocytes, in particular neutrophils, is a factor that raises concern and debate because of their known pro-inflammatory cytokine activities and the release of MMP’s, potentially escalating any early onset inflammatory response in injured hard and soft tissues [73]. The neutrophil specific initial pro-inflammatory actions contributed to a certain level of preoccupation among practitioners regarding their presence in PRP injectates. This concern may be exaggerated, however, considering the article by Jones et al. demonstrating neutrophil’s ultimate role in resolution of inflammation via multiple biological pathways [71]. Consequently, most physicians, and the literature, termed PRP products containing neutrophils LR-PRP, without specifying which leukocytic cells are concentrated in this PRP. This speculative thinking regarding leukocyte cellular effector roles in PRP has been focused on the individual white blood cell type functions, rather than their combined platelet-leukocyte interactions in shaping local and systemic immune responses.

Platelet activation discharges platelet granular storages, with the release of phospholipids from destroyed platelet membranes [109]. These trans-cellular metabolic pathways involve the generation of arachidonic acid-derived lipid mediators from platelet phospholipids, capable of modulating inflammation [110]. The effect of activated platelets on neutrophils allows neutrophils to take up arachidonic acid which was released by platelet membrane phospholipids. This uptake by neutrophils is evocative as they can convert platelet-derived arachidonic acid lipids into various leukotrienes (LTs) and other molecules [111]. LTs function as effective chemotactic signals to enlist immune cells [112]. However, platelets connected to neutrophils that are producing LTs can rapidly consume these inflammatory arachidonic lipid mediators and convert them to lipoxins (LXs) [113]. Lipoxins are metabolites in the arachidonic acid pathway, as described by Hamberg and Samuelsson [114], and are very potent anti-inflammatory molecules that play important roles in limiting pro-inflammatory neutrophil activation, preventing their migration into tissues, and in driving the resolution of inflammation [110,115]. It is important to note that platelets lack the ability to synthesize lipoxins without the LTs produced by neutrophils [116]. This switch from initial generation of pro-inflammatory lipid mediators into ultimately anti-inflammatory lipid mediators in neutrophils is thought to prevent further neutrophil recruitment and inflammatory activation while simultaneously activating resolution pathways that can accelerate tissue healing [117].

C-PRP bioformulations with a significant platelet dose and monocyte yield will mimic the natural activity of Th-2 cells to produce anti-inflammatory IL-4 at the treatment site. This is an important outcome, as IL-4 will support macrophages to convert to MP-2, the regenerative phenotype, while IFN-γ alters macrophages to the inflammatory MP-1 type, in a dose and time dependency. Likewise, IL-4 activation stimulates MP-2 to differentiate Treg cells to Th2 cells, potentially producing additional IL-4 [118].

Interestingly, in response to tissue-derived biologics, Th cells lead macrophage phenotypes to pro-regenerative phenotypes, in an IL-4-dependent manner [119]. In this context, PRP platelet molecules can instigate the polarization and differentiation of macrophage subtypes through the production of various macrophage stages [120,121] and potentially modulate macrophage crosstalk with other immune cells, e.g., nuclear factor kappa light chain enhancer of activated B cells (NF-kB) pathways in synovial cells and chondrocytes [122]. In C-PRP, the high dose of platelets and monocytes signify a unique biological treatment potential. Both concentrated cell types release an abundance of proteins and molecules to stimulate the complex cell-cell interactions in immunomodulatory mechanisms related to OA and tendinopathies [123]. Regrettably, the inflammation resolution functions of leukocytes in injected LR-PRP are generally not thought of, nor the leukocyte-platelet interaction functions, or local tissue effects. Additionally, the locally present leukocyte populations that have been chemotactically recruited from the circulation into treated tissues in response to a C-PRP treatments are also all activated [124]. There appears to be a clear difference between platelet primed but not activated leukocytes. Importantly, non-activated leukocytes when combined with platelets, release mediators that prevent the further recruitment of activated cells and they can suppress inflammation [125,126]. The potential of PRP orthobiological products where platelets are primed, but not activated with (specific) leukocytes, requires more in-depth study at a physiological level. Further clinical research should be directed towards the efficacy in clinical outcomes when LR-PRP products are used to reduce pain and improve function. In the clinical literature, non-conclusive benefits of LP-PRP compared to LR-PRP in knee OA orthobiological PRP applications have been published [76,127,128,129]. Recently, Abbas and co-workers performed a network meta-analysis reviewing the effects of LP-PRP and LR-PRP. They analyzed 23 studies with 2260 patients included, and a10-month follow up period [130]. They found no significant (*p* < 0.05) difference in all outcome measures, with no statistical difference in local adverse reactions between LP-PRP and LR-PRP. Moreover, in soft tissue pathologies, the benefits following LR-PRP applications have been recently demonstrated without any significant adverse events related to the use of a full buffy coat PRP product [131,132].

## 6. Transforming PRP Biology in Orthobiological Applications

The musculoskeletal system provides form, stability, and movement to the human body. It consists of various systems (bones, muscles, tendons, ligaments, joints, cartilage, fascia, and other connective tissue) each with multiple different tissue structures and distinctive molecular and cellular compositions. Importantly, the different tissue structures must maintain a mechanical and biological balance, supporting, generating, and responding to physical forces and various stresses. The MSK system enables humans to move using their neural, myofascial, myotendinous, and skeletal systems, each of which has characteristic functions and biomechanical properties. In the event of functional system instabilities, lesions, trauma, or degenerative processes, orthobiological therapies can be indicated to treat these various structures. The precise administration of specific concentrated autologous cells and orthobiologic substances into targeted pathological local microenvironments leads to consistent, reproducible, and augmented functional patient outcomes. Specifically, as discussed extensively in this review, a particularly convincing and effective tool utilized in interventional orthobiological tissue repair is the administration of a high quality autologous PRP formulation, consisting of high dose platelets and multi-cellular leukocyte populations, specific to the tissue being treated.

A so-called “one-size-fits-all” and basic PRP preparation approach should be abandoned as a standard for all types of MSK pathologies. Neglecting the ability to prepare tissue-specific and personalized orthobiological PRP products regarding platelet dose and customized bioformulations leads to sub-optimal patient care, limiting immunomodulatory and angiogenetic tissue responses, and therefore tissue repair. Furthermore, an “all-purpose, suboptimal” approach is a likely explanation for the variable and conflicting data published regarding PRP therapies.

### 6.1. Transitioning to Evolutionary Medicine for Orthobiological Pathologies

We propose the implementation of changes based on the principles of evolutionary medicine to better understand and optimize orthobiological treatments [10]. Evolutionary medicine is recognized as a core concept in biology to better comprehend biological disease states and thus the implementation of rational treatment options to those pathologies. Evolutionary medicine is a rapidly developing field covering many topics in health, disease, aging, epigenetics, nutrigenomics/pharmacogenomics, and treatment options [133]. In this respect, we suggest considering PRP as an autologous, personalized, compounded medication with variable (platelet) dosing options and variable leukocyte constituents (lymphocytes, neutrophils, and monocytes), where each variable can be considered as a potential outcome driver. Adjusting these PRP variables can invigorate the potential for significant immunomodulatory activities, (neo)angiogenesis, tissue repair, regenerative pathways and ultimately tissue healing. Additionally, a totally integrated systems approach to treatment of MSK pathologies should be embraced.

#### 6.1.1. Case Presentation

Here, we present a case to illustrate the clinical feasibility, based on the principles of evolutionary medicine, of treating a patient (PE) with multiple knee pathologies in the same treatment session. From a unit of whole blood, tissue type and pathology specific PRP injectates were prepared, with emphasis on platelet dosing and bioformulations, abandoning the concept of all-purpose PRP preparations.

In accordance with the MRI findings, listed here below, a personalized treatment plan was designed to treat the various knee joint pathologies utilizing an expanded arsenal of therapeutic PRP injections. A personalized multitargeted interventional injection treatment plan was created as shown in Table 2. The objectives of the treatment plan were not only relief of symptoms (pain), but also focused on tissue repair strategies since the knee joint is an essential part of the MSK system. Informed consent was obtained for peripheral blood draw, ultrasound guided knee joint aspiration, and PRP cell-based therapy with non-activated autologous NP-PRP and NR-PRP to inject the left median collateral ligament (MCL), anterior cruciate ligament (ACL), medial meniscus, meniscal coronary ligaments, meniscal capsular junction, periarticular articular capsule.

Established evidence-based data shaped the plan to use a 2-step algorithm, including both NP-PRP for some injection sites and NR-PRP for others.

##### Pathology Findings following MRI

Complex posterior horn meniscus tear extending from articular surface to peripheral capsular margin.MCL intact; medially bowed with stripping of proximal tibial attachment, and diffusely thickened fibers.Mild OA medial compartment with joint space narrowing, and small osteophytes.Moderate diffuse chondromalacia anterior medial condyle.Mucoid degenerative, thickened, ACL.

##### Motivation to Use NP- and NR-PRP Formulations

For the intra-articular injection, NP-PRP was prepared, based on the positive clinical outcomes of a systematic review. According to the authors, leukocyte-depleted PRP provides superior results [14]. The long-term positive clinical outcomes reported by Bansal et al. regarding pain and function encouraged us to use NP-PRP platelet dosages similar to their reporting, thereby exceeding the platelet doses used in previous publications showing no positive results [6,7]. Likewise, a systematic review conducted by Muchedzi and Roberts also found that PRP can significantly reduce pain in knee osteoarthritis [83].

All knee soft tissue injections were executed with NR-PRP. Regarding meniscus tears, a systematic review and meta-analysis revealed that PRP significantly reduced the failure rate of meniscus repair as well as improve pain control. Additionally, significant improvement in outcomes was observed in studies that used higher concentrations of PRP [131]. In a blinded randomized controlled trial, the effectiveness and safety of percutaneous intra-meniscal PRP application was investigated to complement repair of a chronic meniscal lesion. Intra-meniscal NR-PRP was injected, following a double-spin preparation procedure of 120 mL of whole blood. The authors concluded that the procedure resulted in significant improvements in the rate of chronic meniscal tear healing and a decreased incidence for future arthroscopy procedures [134]. With regard to dosing, we adopted the recommendation of Hutchinson and Rodeo, recently suggesting a minimum of 1,000,000 platelets/μL to treat isolated meniscus tears [135].

As for the ligament injections, we chose a NR-PRP bioformulation based on the extensive work and long-term follow-up of Podesta et al. [136]. This study used LR-PRP to treat elbow partial ulnar collateral ligament tears. Long-term follow-up (average 70 weeks) showed that NR-PRP is an effective non-surgical treatment option for ligament tears. These positive effects of NR-PRP on treating ligament pathologies was later demonstrated by Deal and his colleagues [137].

The mucoid degenerative ACL was treated with NR-PRP injections, delivered to the intercondylar notch and tibia insertion, even though arthroscopic debridement is usually recommended [138]. The rationale to use NR-PRP injections to treat the mucoid ACL was based on the fact that, apart from releasing platelet constituents, neutrophils, monocytes, and other NR-PRP immune cells produce an inflammation response causing ligament tissue debris and damaged cells to be ingested and removed, which leads to ligament matrix turnover [139].

##### Clinical Procedure

The procedure was started, after obtaining informed consent, with a 17 G butterfly needle inserted in the antecubital vein, to facilitate a venous blood draw to determine, at point-of-care, a baseline complete blood count (CBC), according to good laboratory practice guidelines using a calibrated hemo-analyzer (Beckman Coulter DxH 500-5-part differential, Brea, CA, USA). Following CBC analysis, the platelet dose, bioformulation, and injection volumes were calculated for each tissue structure being treated. Second, based on the pre-procedure CBC baseline platelet concentration and the 85% platelet capture of the PRP device being used. It was calculated that the total predonation whole blood volume should be 108 mL (excluding a total of 12 mL of sodium citrate anticoagulant), collected in two 60 mL syringes, to produce adequate final volumes and platelet dose of NR -and NP-PRP following a 2-spin centrifugation procedure, as shown in Figure 3. Subsequently, the two different PRP bioformulations were prepared in less than 10 min, following the instructions for use of the manufacturer. The final prepared PRP products were aseptically transferred in 4 injection syringes, based on the calculated volumes for the various tissue structure injections, as illustrated in Figure 4. CBC analysis was performed in the final NR- and NP-PRP products.

A total of 7 articular and soft tissue injections and a synovial fluid extraction were performed under direct ultrasound imaging, using 4 skin entry points. The injected tissue structures and the injectate specific biological profiles are presented in Figure 4.

During the PRP preparation procedures, an extensive ultrasound scanning protocol of the right knee tissue joint structures preceded two regional anesthesia blocks (Lidocaine 0.5%) of the adductor canal (saphenous nerve block) and genicular nerve, a motor-sparing technique that anesthetizes the sensory terminal branches that innervate the knee joint.

Succeeding the preparation of the injection syringes, a 25 G 1.5-inch needle was inserted in all target areas, depositing 0.2 mL of local anesthetic (Ropivacaine 0.5%, to minimize cytotoxic effects of the PRP cells) in the target areas. Thereafter, the needle was left in-situ and the PRP administered, as indicated in Figure 4. The injection volumes of the various tissue structures were meticulously recorded to precisely calculate and evaluate the delivered platelet dose per tissue structure. The entire treatment procedure went without any complications, or adverse events. Post procedure, a personally fitted and adjusted uni-compartmental knee brace with biomechanical properties (Össur One X Medial Unloader Brace^®^, Foothill Ranch, CA, USA) was put on to provide pain relief and support in functional activities for 10 weeks, at all times.

##### Post Treatment Observations

Just prior to submitting this manuscript, at 11 weeks post treatment the right knee was re-evaluated by the same provider (L.P.) using the same ultrasound device. In Figure 5, a comparison image is shown from the medial meniscus and MCL. In 5A, a re-evaluation image is shown: the anechoic defect affecting the red and red/white zone of the meniscus, as noted on the ultrasound image in 5B, is barely visible; this is compatible with a tissue regenerative repair process of the meniscus tear following NR-PRP administration in the meniscus and the femoral and tibial meniscus ligaments. Furthermore, the International Knee Documentation Committee (IKDC) score and the Knee Injury and Osteoarthritis Score (KOOS) improved. At 11 weeks post treatment, both scores increased, 48% and 53%, respectively. The visual analog scale for pain decreased from 6 to 0 in the same period.

## 7. Discussion

There is a global unfulfilled demand to treat a wide range of MSK-disorders employing interventional, non-surgical, innovative orthobiological procedures, using autologous orthobiological cellular preparations [140]. Patients with a variety of compelling clinical problems, who have previously shown limited response to medications, rehabilitation, surgery, or even joint replacement surgery, might benefit from these biological therapies. Orthobiologics have the capacity to facilitate and expedite natural tissue repair mechanisms, specifically in tissues where natural healing is disrupted or incomplete, where degenerative biological processes are breaking down tissues, where blood flow and tissue repair is at risk, and ultimately where structural integrity is degenerative, compromised, or broken.

PRP therapies have become an important autologous ortho-biological therapeutic treatment tool. PRP technology, including device specific preparations, and orthobiological treatment protocols, has evolved immensely over the past 20 years. The original biological rationale for the clinical use of PRP includes the local delivery of the intracellular platelet vesicles, containing a wealth of growth factor proteins and many other molecules. Over the last decades a better apprehension of the functions of other biological components in the platelet proteome that affect PRP-treatment outcomes have emerged [16]. This is particularly true as we learned of PRP’s effectiveness in stimulating cell proliferation and differentiation, guiding regenerative processes, immunomodulation, angiogenesis, pain killing, tissue repair, and restoration of function [68,141]. Generally, PRP therapies lead to promising results as they assist the body’s innate healing abilities in establishing new functional tissues in place of degenerative or defective ones. Unfortunately, the literature also reports on studies showing no positive or negative patient outcomes. Arguments to explain these inconsistent results are multidimensional: 1. the PRP device market is filled with many different types of devices that are not proven to be effective in certain pathologies; 2. there is no consensus regarding PRP quality standards; 3. validated PRP preparation guidelines are lacking; 4. many treatment sites have opened; 5. there is a lack of certified regenerative orthopedic educational programs. In order to resolve these inconsistent and negative outcomes in MSK conditions, as well as the associated unenthusiastic recommendations, we proposed some novel concepts and redesigned orthobiological preparations and treatment plans.

This transformative approach includes the following recommendations: 1. optimization of high cell yield PRP preparation techniques; 2. implementation of integrated systems treatment strategies, and 3. the use of different PRP formulations to treat different tissue types and conditions in the same procedure.

## 8. Conclusions

It is imperative that physicians remain aware of the large variability of orthobiologic PRP devices that are currently available on the market. Rather than focusing on a single type of PRP for all patients and conditions, we need to avoid an overly simplistic “one size fits all” approach. Orthobiologic-oriented bench research must be intertwined with clinical studies aiming to demonstrate greater efficacy and better clinical outcomes than currently accepted traditional treatments. Moreover, more platelet dosing studies should be conducted to provide clear information on TDP strategies for different tissues, showing consistent positive results.

We are convinced that a multi-dimensional, agile, and tissue-specific repair approach has the potential to optimize the full complexity of the innate healing response via biological cell signaling to functionally restore orthopedic structures and microenvironments. Future clinical studies should focus on the detailed effects of platelet dosing and bioformulations applied to different pathological tissues, to solidify PRP orthobiological strategies for the advancement of therapy outcomes in musculoskeletal pathologies.

## Figures and Tables

**Figure 1 biomedicines-10-02933-f001:**
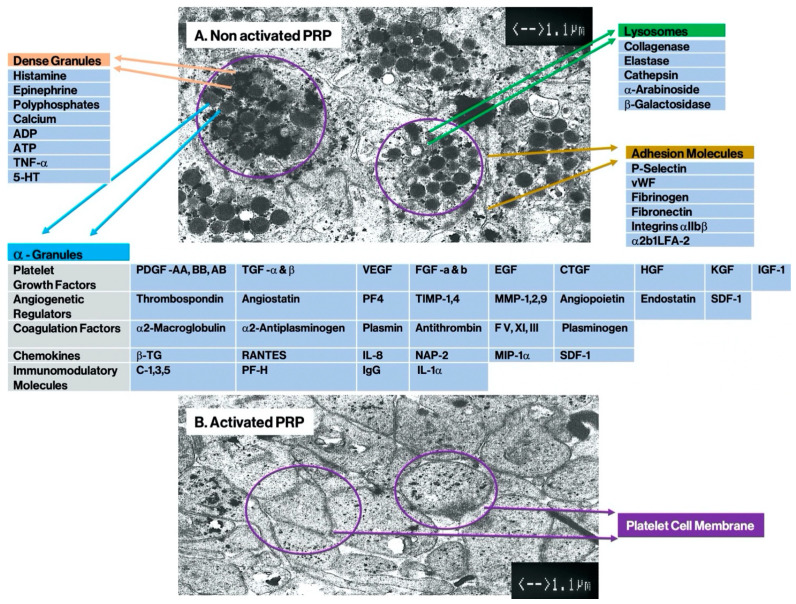
Platelet granular storages and their content in non-activated platelets. Electron microscopic scanning pictures of a cluster of platelets, at original magnification ×10,000. In (**A**), non-activated platelets from a PRP vial (PE) are shown. The ill-defined grey bodies represent the α-granules, approximately 50–80/platelet. The opaque round bodies are the dense granules, ranging from 0–30/platelet, and grey string-like bodies are lysosomes [24]. Thin lines signify platelet cell membranes. In (**B**), disrupted cell membranes are visible as activated platelets have released their granular contents (Abbreviations: ADP: adenosine diphosphate; ATP: adenosine triphosphate; TNF: tumor necrosis factor; 5-HT: serotonin; vWF: von Willebrand Factor; PDGF: platelet-derived growth factors; TGF: transforming growth factor; VEGF: vascular endothelial growth factor; FGF: fibroblast growth factor EGF: epidermal growth factor; CTGF: connective tissue growth factor; HGF: hepatocyte growth factor; KGF: keratinocyte growth factor; IGF: insulin-like growth factor; TIMP: Tissue inhibitor of metalloproteinase; MMP: matrix metalloproteases; F: coagulation factor; β-TG: beta-Thromboglobulin; NAP: neutrophil-activating peptide; MIP: Macrophage inflammatory protein; SDF: stromal cell derived factor; C: complement protein; Ig: Immunoglobulin). Adapted and modified from Everts et al. [3].

**Figure 2 biomedicines-10-02933-f002:**
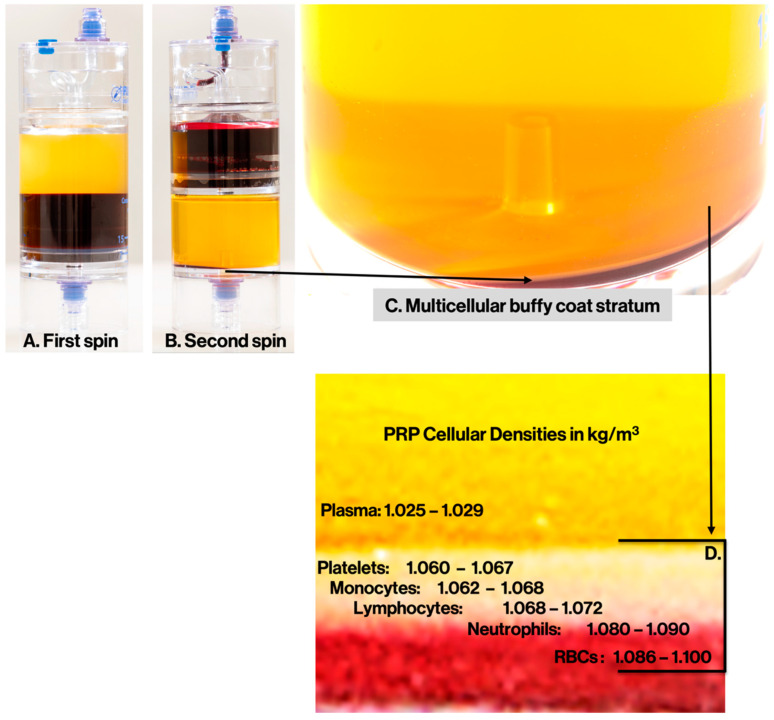
Multicellular Density Separation after a Double-spin PRP Procedure. Graphical presentation of the cellular gravitational density separation using a 60 mL PurePRP-SP^®^ device (Used with permission from EmCyte Corporation, Fort Myers, FL, USA). After the first spin (**A**), the whole blood is sequestered in a top layer (plasma fraction), an intermediate thin layer (multicomponent buffy coat, including platelets and leukocytes), and a bottom layer consisting of RBCs. After the second spin (**B**), all cells are concentrated at the bottom of the second compartment of the PRP device. In (**C**), after a magnification of the second compartment after the second spin, the multicellular buffy coat stratum (grey layer on top of the RBCs). Further enhancement, (**D**) picture, indicates the representation following cellular gravitational density separation, according to the specific densities of the individual cells [94]. Caveat: note the overlap in the range of cellular densities of the various cell types.

**Figure 3 biomedicines-10-02933-f003:**
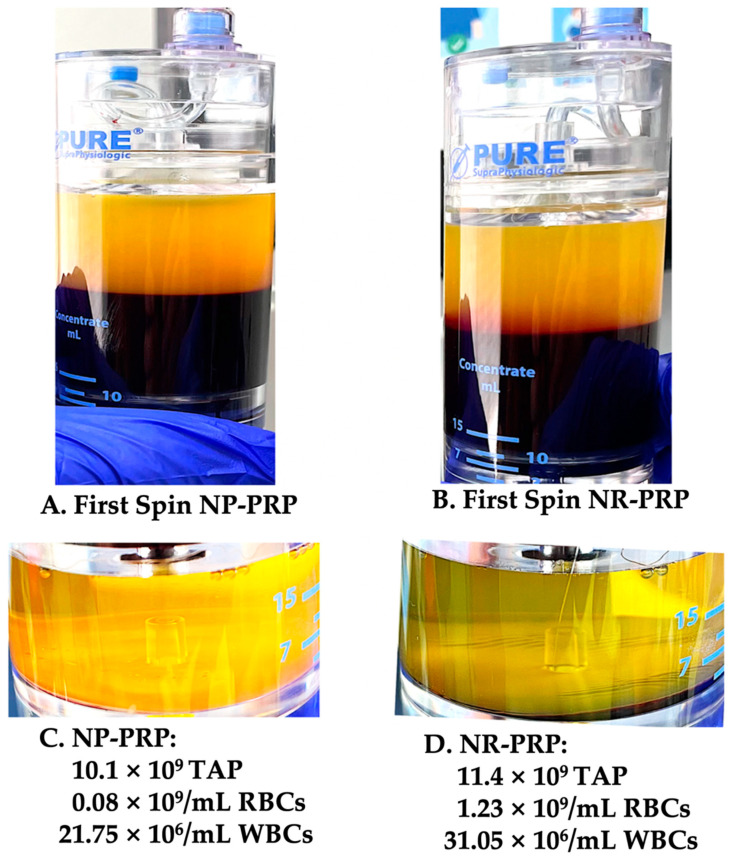
Visual aspects after the First and Second spin of NP-PRP and NR-PRP Preparations. After the first centrifugation spin, no differences occur for NP-PRP (**A**) and NR-PRP first phase preparation (**B**). During this cycle, the unit of whole blood is sequestered in a platelet poor plasma (PPP) fraction, thin buffy coat layer, and a packed RBC layer. After the second spin, the plasma in the second device chamber is removed until the remaining volume in each device is in accordance with the total calculated treatment volumes for both bioformulations. Thereafter, the cells at the bottom of the chamber are resuspended with the remaining plasma. Laboratory data present the differences between the NP-PRP (**C**) and NR-PRP (**D**) fractions. Note, the broader RBC layer in the NR-PRP vial (**D**) when compared to the NP-PRP preparation (**C**). Furthermore, the NR-PRP buffy coat stratum on top of the RBCs is more noticeable (PurePRP-SP^®^ device, used with permission from EmCyte Corporation, Fort Myers FL, USA) (Abbreviation: TAP: total available platelets).

**Figure 4 biomedicines-10-02933-f004:**
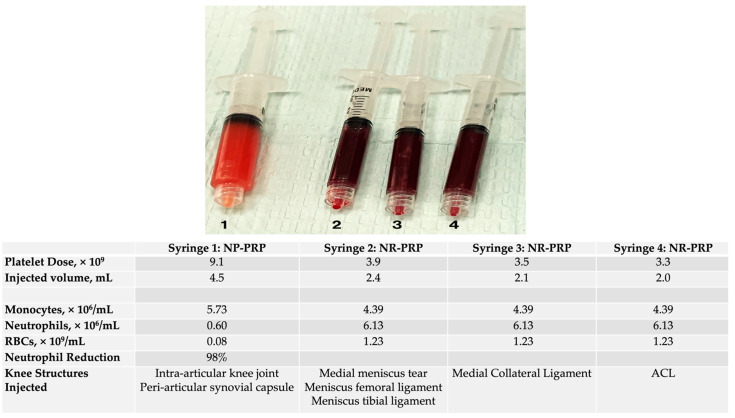
Description of Dosing and Bioformulations of the various injected Tissue Structures. Both, NP- and NR-PRP prepared volumes were transferred in smaller application syringes. The volumes in these syringes are corresponding with the calculations following the algorithm used, safeguarding platelet dose and bioformulations to treat the various tissue structures based of the pre-procedure treatment plan.

**Figure 5 biomedicines-10-02933-f005:**
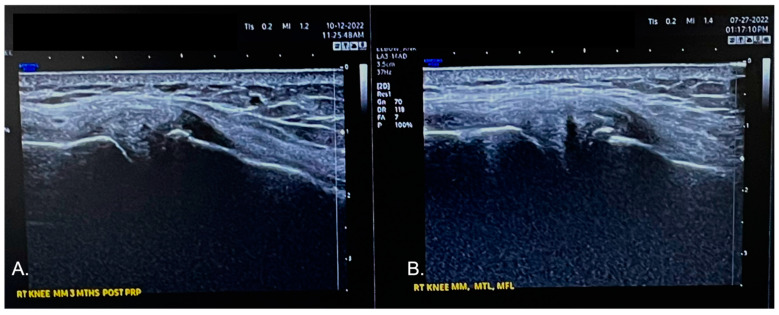
A comparative image of the right knee showing medial meniscus and MCL. In situation (**A**), 11 weeks after the procedure, the medial meniscus is visible following an intra-meniscal NR-PRP injection. In situation (**B**), the pre-treatment condition of the meniscus is visualized.

**Table 1 biomedicines-10-02933-t001:** Platelet concentration and platelet dosing discrepancies as quality indicators for injected PRP in same patient model.

	Angel^®^	GPSIII^®^	PurePRPII^®^-A	PRF	Regenkit^®^-A
PLT increase from Baseline	4.8	4.2	6.6	1.9	0.7
Platelet Concentration (×10^6^/mL)	856	754	1175	338	125
PRP volume (mL)	3	6	7	2	5
TDP (×10^6^/mL)	2568	4524	8225	677	623

Variances in PRP specimen, prepared with 5 different commercial products (Angel^®^: Arthrex, Naples, FL, USA; GPS III^®^: Zimmer Biomet, Warsaw, IN, USA; PurePRPII^®^-A: EmCyte Corporation, Fort Myers, FL, USA (protocol A preparation); Regenkit^®^: Mont-sur-Lausanne, Switzerland). In this same donor model (N = 12), with age ranging between 45–60 years old, 325 mL of whole blood was donated at all times to prepare PRP according to the manufacturer’s instructions for use. The average whole blood platelet count was 178 (×10^6^/mL). The data are mean values. Unpublished data (Abbreviations: PLT: platelet).

**Table 2 biomedicines-10-02933-t002:** Blueprint of the Personalized PRP Treatment Plan Right Knee.

Baseline CBC Data	Platelets Leukocyte Differentiation
Ultrasound scanning knee joint tissue structures	Joint intraarticular Tendons and Ligaments Menisci
Regional anesthesia	Right adductor canal nerve block Right geniculate nerve block
Peripheral anticubal vein whole blood aspiration	17G NIPRO fistula needle
Calculated predonation volume:	108 mL in two 60 mL syringes Each syringe holds 6 mL of sodium citrate
2-Spin PRP preparations	EmCyte PurePRP-SP^®^: two 60 mL concentration devices
NR-PRP: anticipated preparation volume	7.0 mL (6.5 treatments, and 0.5 lab)
NP-PRP: anticipated preparation volume	5.0 mL (4.5 treatment, and 0.5 lab)
Laboratory validation PRP preparations	Beckman Coulter DxH500, 5-part diff
PRP tissue injection structures	NP-PRP: Intra-articular knee joint NP-PRP: Periarticular capsule NR-PRP: Medial meniscus tear NR-PRP: Meniscocapsular junction NR-PRP: Meniscus femoral ligament NR-PRP: Meniscus tibial ligament NR-PRP: MCL complete NR-PRP: ACL
Heat pack protocol	
Knee brace protocol	Össur One X Medial Unloader Brace^®^

## Data Availability

The medical information is at the office of L.P.

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
