# Peer review of "Modifying Orthobiological PRP Therapies Are Imperative for the Advancement of Treatment Outcomes in Musculoskeletal Pathologies"

_biomedicines, 2022, doi:10.3390/biomedicines10112933_

Round 1
Reviewer 1 Report
This manuscript attempted to introduce the principles of "evolutionary biology" to improve the clinical effectiveness of PRP formulations.
A major limitations of the manuscript was the authors' attempt to calculate the PRP dose for different tissues through an "algorithm" so that "The injection volumes of the various tissue structures were meticulously recorded to precisely calculate the delivered platelet dose per tissue structure", but I have concerns about how well this can be implemented, as in the case of the appropriate platelet injection volume required for each treatment site was not described. The authors themselves stated that "further studies are warranted to understand what TDP dosing is optimal for specific tissue types", More detail is needed in the limitations section regarding this.
A few other questions:
1. The authors discussed the different leukocyte types including lymphocytes, monocytes and neutrophils, but only two different biological agents, neutrophil poor PRP (NP-PRP), neutrophil rich PRP (NR-PRP) have been achieved. Considering the different roles of different white blood cells mentioned in the manuscript, is it possible to achieve PRP containing different doses of monocytes and lymphocytes?
2. In the case the authors used NP-PRP and NR-PRP for different site injections but did not state the reason for this classification, the manuscript mentioned "leukocyte rich PRP is also frequently cited as the orthobiologic of choice in chronic tendinopathy treatments" (page9) may be a reason? A more detailed explanation would be nice.
3. The author mentioned “the total donated blood volume” responsible for the heterogeneity of PRP preparations in MSK applications, which I don't quite understand (page 11).
4. “the optimal platelet dose to promote angiogenesis was 1.5 x 106 platelets/μL” (page14)and “it is fair to assume that PRP preparations with a high platelet dose contribute to significant positive angiogenetic effects, when compared to lower platelet products with less than 1.5 x 106 platelets/μL” (page14)seem to be somewhat contradictory, since a dose above the optimal dose is not necessarily better than a dose below the optimal dose.
5. In the case presentation, the MRI findings of various degrees of knee joint pathologies were listed, and it would be nice to add the MRI findings of the follow-up after PRP treatment.
Reviewer 2 Report
The manuscript submitted by Everts and colleagues comprehensively summarizes the published information on orthobiological PRP therapies applyed for treatment of musculoskeletal disorders, osteoarthritis, cartilage defects, tendinopathies and other soft tissue maladies. The review article is clear and well structured. It will attract the interest of researchers working in the field. Important and intriguing part of the manuscript is the discussion and recommendation for transformations in routinely accepted orthobiological PRP preparations and treatment protocols based on the principles of evolutionary medicine. However, a major drawback of this work is the unification of two scientific terms, which are completely different - evolutionalry biology and evolutionary medicine. It is inappropriate to use the term "evolutionary biology" as a synonym of "evolutionary medicine". Moreover, one of the keywords listed by the authors is “evolutionary biology”, which is misleading. Evolutionary biology is a biological subdiscipline that aims to clarify the origin of life, the diversification and adaptation of life forms over time. Thus, the authors should carefully revise the text, as well as the title, and replace “evolutionary biology” with more appropriate terms. In line with this topic, subsection 6.1.1. should be improved. The authors should motivate and improve the description of the personalized treatment plan. Did the authors observe positive results after the treatment?
Another weakness of the manuscript is the presence of many sentences and subsections that show high similarity with previously published article (https://doi.org/10.3390/ijms21207794).
Other points that need to be addressed are:
11) References should be formatted according to the journal guidelines.
22) The abstract and conclusions are too long. Please revise.
33) Abbreviations should be defined the first time they appear in each of three sections: the abstract; the main text; the first figure or table.
44) Figure 1, A – the picture has already been published. The authors should include reference or modify the figure.
55) The phrase “release of platelet lysosomes” is not appropriate. Please replace with “release of lysosomal content”.
66) Duplicated sentence: L491-493 and L495-497.
Round 2
Reviewer 2 Report
The manuscript has been improved but it needs additional revision. The change of "evolutionary" with "revolutionary" is not appropriate. The manuscript suggests personalized changes in orthobiological PRP therapies. Changes that cannot be specified as revolutionary. The authors need to revise again the title and the main text and replace the phrase "revolutionary" with words that better describe the essence of the suggested changes.
Subsection 6.1.1. has been expanded with methodological description. However, a clear motivation of the choice of treatment plan has not been presented in the begging of the subsection.
The parts with detected autoplagiarism have not been revised. It is not acceptable to publish again text that is part of another article. The authors should modify subsections 2.1.4., 3.3.,3.3.1.,3.3.2.,3.3.3. and 3.4.
Author Response
Thank you for your second review report.
We changed the title to: "Modifying Orthobiological PRP Therapies are Imperative for the Advancement of Treatment Outcomes in Musculoskeletal Pathologies".
We removed revolutionary in line 66.
We modified lines 870-874, eliminating revolutionary
The Case presentation at 6.1.1 has been modified to accommodate your request for motivation to use NR- and NP-PRP. We added now 6.1.1.2 Motivation to use NP- and NR-PRP.
6.1.1.3 is now titled Clinical procedure, followed by 6.1.1.4 Post treatment observations.
6.1.1.1 is moved upwards below 6.1.1 for more clarity and easier reading, according to authors and hope you agree.
A few additional references were added to section 6.1.1.2 to complete the motivating articles regarding formulations.
Your comments regarding detected auto plagiarism have been addressed and these and other subsections have been corrected, with letter color blue.
We hope to have answered and responded correctly to your questions and comments.
Respectfully, on behalf of the authors.
P. Everts
***Author's Response to Review Report 1 was in the attachment.

Round 3
Reviewer 2 Report
The authors have now addressed all my comments and suggestions.
It was difficult to review all corrections in the manuscript because a significant part of them have not been marked (including the changes in the title).
My final remark is the Conclusions section, which is too long and could be divided in two sections - discussion and conclusions.
The references need to be formatted according to the journal guidelines. Although the authors have stated that they have revised the references they still don't correspond to the style required by mdpi.
Author Response
Dear reviewer,
Thank you again for you review and critical review.
Our apologies for missing the track changes in previous version, we focused on changing the entire the text paragraphs.
We followed your advice and added a discussion section and a shorter conclusion section, as paragraph 8.
The references are addressed and a Zotero Style Repository for Biomedicine was installed and refreshed in this manuscript version.
We hope to have completed all your review suggestions in this uploaded version.
Again, thank you for supporting our manuscript. We are convinced that with your comments a bette manuscript has bee realized.
On behalf of the authors,
Peter Everts
